# Systems to Monitor the Individual Feeding and Drinking Behaviors of Growing Pigs Based on Machine Vision

**Yanrong Zhuang [1,2], Kang Zhou [1,2], Zhenyu Zhou [1,2], Hengyi Ji [1,2] and Guanghui Teng [1,2,3,*]**

1   College of Water Resources & Civil Engineering, China Agricultural University, Beijing 100083, China
2   Key Laboratory of Agricultural Engineering in Structure and Environment, Ministry of Agriculture and Rural Affairs, Beijing 100083, China
3   Beijing Engineering Research Center on Animal Healthy Environment, Beijing 100083, China
*   Correspondence: futong@cau.edu.cn

**Abstract:** Feeding and drinking behaviors are important in pig breeding. Although many methods have been developed to monitor them, most are too expensive for pig research, and some vision-based methods have not been integrated into equipment or systems. In this study, two systems were designed to monitor pigs' feeding and drinking behaviors, which could reduce the impact of the image background. Moreover, three convolutional neural network (CNN) algorithms, VGG19, Xception, and MobileNetV2, were used to build recognition models for feeding and drinking behaviors. The models trained by MobileNetV2 had the best performance, with the recall rate higher than 97% in recognizing pigs, and low mean square error (RMSE) and mean absolute error (MAE) in estimating feeding (RMSE = 0.58 s, MAE = 0.21 s) and drinking durations (RMSE = 0.60 s, MAE = 0.12 s). In addition, the two best models trained by MobileNetV2 were combined with the LabVIEW software development platform, and a new software to monitor the feeding and drinking behaviors of pigs was built that can automatically recognize pigs and estimate their feeding and drinking durations. The system designed in this study can be applied to behavioral recognition in pig production.

**Keywords:** feeding behavior; drinking behavior; CNN; MobileNetV2; LabVIEW

## 1. Introduction

Monitoring the behaviors of pigs can reflect their health status timely [1]. Feeding and drinking behaviors are two behaviors with a strong relationship [2], and there is a positive correlation between feed and water intake [3]. Monitoring of feeding and drinking behaviors can help users determine pigs' feed intake [1], carry out relevant nutritional studies [4], and keep up with the environmental condition in pen, health, and social stressors of pigs [5].

Traditional methods of monitoring the feeding and drinking behaviors of pigs require a large number of work hours and might stress the pigs [6]. In some research, to improve the efficiency of behavior monitoring, radio frequency identification (RFID), which can accurately identify pigs, was used to monitor feeding and drinking behaviors [6–10]. In addition, an electronic feeding station (EFS) or feed intake recording equipment (FIRE), integrated with many sensors including RFID and the others mentioned above, can monitor the behaviors of pigs and include many other parameters in addition to feeding and drinking behaviors [11,12]. However, because of the high cost of the EFS and FIRE, they are usually used in research on sows and boars, but hardly ever for growing pigs. Moreover, the EFS, FIRE, and RFID systems need to pierce pigs' ears to place the tags, which causes stress.

With the development of computer science, machine vision methods are widely used in the recognition of pigs' feeding behavior [13], drinking behavior [14], identity [15], and weight [16]. The deep learning (DL) method, especially the convolutional neural network (CNN) method, a widely used method in machine vision, has developed rapidly in recent

years. Because of the fast processing speed and the end-to-end manner, CNNs are used in many areas including animal science. Many studies on recognizing pigs' identity and feeding and drinking behaviors [2,13,15,17–20] used the CNN method and obtained good results. However, these methods remained in the model establishment stage, did not form equipment or systems for recognition, are not suitable for group-housed pigs, cannot be applied in real time, are limited by usage scenarios (pen size, feeding density, etc.), some are lack of individual recognition or have low individual recognition performance, and some research based on object detection algorithm required complex program compilation. Therefore, it is necessary to build a system that is more adaptable and can be used in pig farms.

In this research, a system that can recognize pigs' individual feeding and drinking behaviors based on the machine vision method was established. The main objectives were to: (1) design a system, which includes equipment to monitor pigs' individual feeding and drinking behaviors; (2) establish models based on CNNs, which can be used to recognize pigs' identity, feeding behavior, and drinking behavior; and (3) test the performance and integrate them into a new software.

## 2. Materials and Methods

### 2.1. Design of the Acquisition and Recognition System

The acquisition and recognition system included the feeding behavior acquisition and recognition system (FARS, Figure 1) and the drinking behavior acquisition and recognition system (DARS, Figure 2). The material of both systems was 304 stainless steel. The size of the FARS was 520 × 360 × 1000 mm, with a 360 × 380 × 200 mm built-in feeder. The camera was installed on the rear plate of the FARS and was 750 mm above the ground. In addition, because of the light-blocking effect of the inner wall, an LED belt was installed on the top. The DARS was a 260 × 130 × 530 mm irregular-shaped drinker; the upper and lower ends were semicircles with a radius of 130 mm. The camera was installed on the top of the DARS, and the distance between the drinker and the camera was 380 mm.

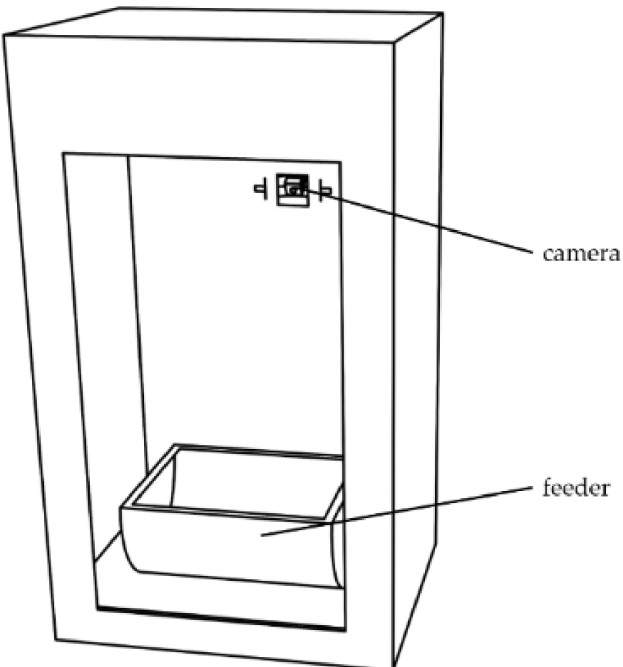

**Figure 1.** Feeding behavior acquisition and recognition system.

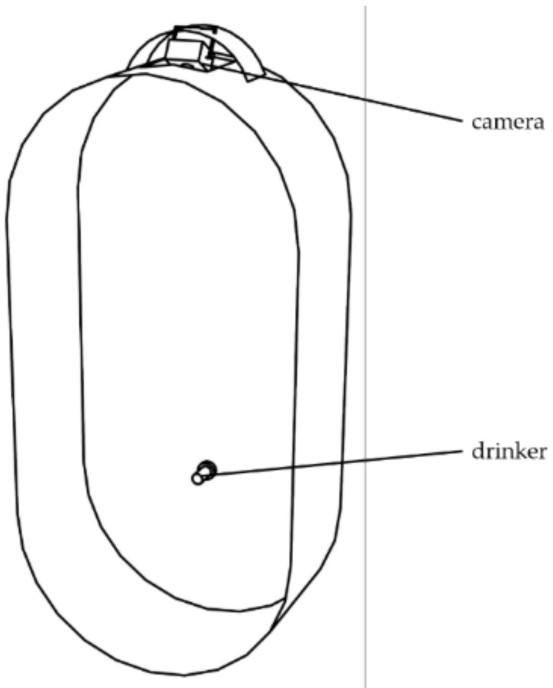

**Figure 2.** Drinking behavior acquisition and recognition system.

### 2.2. Animal and System Layout

This study was conducted in January 2020 at the Ministry of Agriculture Feed Industry Centre Fengning Animal Test Base (Fengning Hebei, China). Six growing pigs (Duroc × Landrace × Yorkshire (DYL)) were monitored in this research, and the pig fodder was delivered at 8 a.m. and 3 p.m. every day. The system was placed on one side of the pen, the FARS was located on the right, and the DARS was placed in the middle (Figure 3). Because a change in the feeder might affect the normal feeding of the pigs, the original feeder was retained [12].

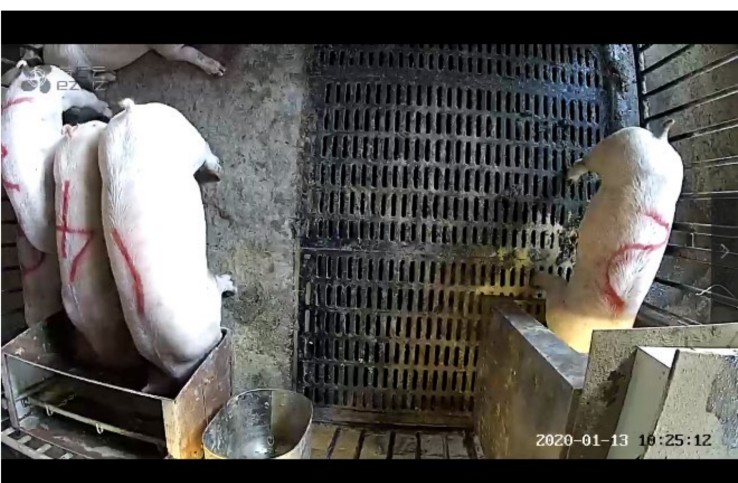

**Figure 3.** Pen and the layout of system.

### 2.3. Programming Language

All code was run on a desktop computer with an Intel i7-10750H CPU, 32 GB RAM, Windows 10 (64 bit), and a NVIDIA GeForce GTX 2070 8 GB GPU.

To train and test the model, the choice of language was significant. The Tensorflow system of Python is a second-generation artificial intelligence learning system developed by Google. Tensorflow is completely open source, has a good development environment,

and is widely used in the field of machine vision research of pigs [15,16], which provides experience for the programming language selection of this study. Therefore, Python was selected as the programming language for model training and verification in this study.

After training and testing, the model should be automatically loaded and applied. Virtual Instrument Engineering Workbench (LabVIEW) is an unusual graphical programming language based on virtual instruments that can easily be used for capturing images and outputting intuitive data. Moreover, the Python integration toolkit LabVIEW can automatically load Python-trained models and generate good display interfaces. Thus, LabVIEW was selected as the language for data acquisition and system integration.

### 2.4. Data Acquisition and Preprocessing

#### 2.4.1. Image Acquisition

In this study, 48,425 images (26,505 from FARS, 21,920 from DARS) were collected from 8:00 to 18:00 per day and named raw data. In addition, 180 consecutive sets of data (50 from FARS, 130 from DARS), which included 4796 images (1809 from FARS, 2987 from DARS), were collected and named test data. The size of the images was 1280 × 720 pixels, and the frequency of taking images was once per second. The cameras used in this research were miniature industrial cameras (LRCP10620_1080P, Shenzhen, China). To better distinguish pigs and facilitate the marking of images, crayons were used to mark the back of the pigs every day. For the cameras placed in the FARS and DARS that could not capture the marks on the pigs' backs, a top-view camera (EZVIZ C3C, Shenzhen, China) was installed to take images of the pen. All images were named according to the time they were taken.

#### 2.4.2. Image Resizing, Cutting, and Marking

Before model training, the image data in this research were resized, cut, marked and cleaned. The resizing and cutting of images made the images suitable for training by different CNN models and saved on calculation costs. The image marking, which included identity marking and behavior marking in this research, was a significant step to teaching the computer the differences between the pigs and their behaviors. The image resizing was based on Python code during training and the model to be used in training; the image cutting was based on the LabVIEW Vision Toolkit; the image marking was based on a comparison of the images from the FARS and DARS to the top camera; the image cleaning was based on the open-source platform of Baidu (https://ai.baidu.com/easydata/, accessed on 7 October 2022), which cleans out blurred and highly approximate data.

The images captured by FARS (Figure 4a, 1280 × 720 pixels) included a lot of useless information, such as other pigs or the inner wall of the FARS, which could affect the performance of the models and make the marking work difficult, so these images were cut to a suitable size (Figure 4b, 640 × 360 pixels). Subsequently, the images of the feeding behavior dataset were reduced from 26,505 to 12,200, and these 12,200 images were marked by the number of pigs. These marked images are shown in Figure 5, with the feeding images of six pigs marked from 1 to 6. Images with no feeding were marked as 0.

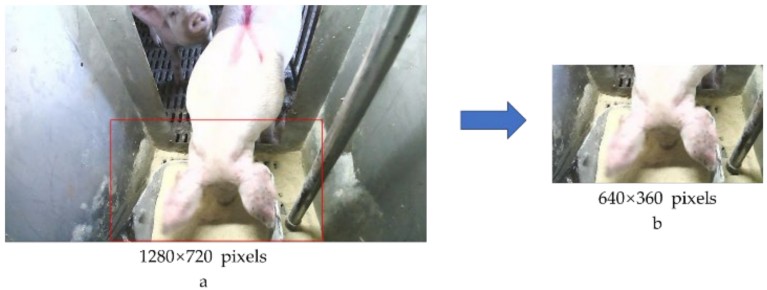

**Figure 4.** Image caught by FARS and image cutting. Raw image (**a**); image after cutting (**b**).

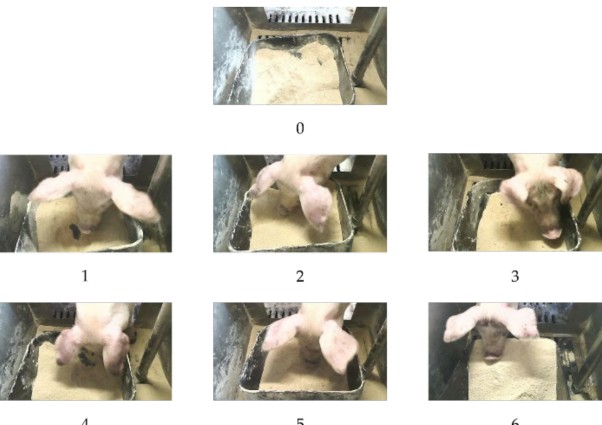

**Figure 5.** Images of individual pigs at FARS. Nonfeeding (**0**); numbers of feeding pigs (**1~6**).

Compared with the images caught by FARS, the images caught by DARS (Figure 6, 1280 × 720 pixels) had little useless information and only needed to be marked and cleaned. After cleaning, the original 21,920 images were reduced to 11,298. The marked images are shown in Figure 6: six pigs that were drinking water were marked from 1 to 6. Images with no drinking were marked as 0.

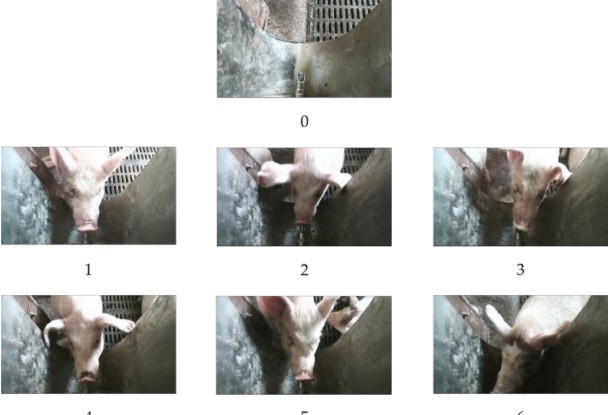

**Figure 6.** Images of individual pigs at DARS. Nondrinking (**0**); numbers of drinking pigs (**1~6**).

After the images were cut and cleaned, in order to be better applied in model training, it was necessary to resize the preprocessed images to 224 × 224 pixels or 299 × 299 pixels for different models, which was the optimal image size for training.

*2.5. CNN Algorithm*

In this study, the raw data were divided into a training dataset and a validation dataset according to a ratio of 4:1. The image data were used as input, and the labels of each image (0–6) were used as output. The models were built based on VGG19, Xception, and MobileNetV2 models. For the training and validation of the three models, the training epochs numbered 30, the training batch size was 32, and the learning rate was 0.0001. All three models had two fully connected layers, and Softmax was selected as the activation function in the output layer. In addition, the sparse categorical cross-entropy was used to determine the loss, and Adam was used as an optimizer. For testing, recall rate was used to evaluate the performance of the three models used in recognizing pigs' individual feeding and drinking behaviors, and the formula was as follows:

$$\text{Recall rate}(i) = \frac{t_{p(i)}}{T(i)} \times 100\%, \tag{1}$$

where $t_{p(i)}$ is the number of results identifying pig $i$ as pig $i$, and $T(i)$ is the total number of images of pig $i$ in the dataset.

Moreover, the root mean square error (RMSE) and mean absolute error (MAE) were used to evaluate the estimation of feeding duration and drinking duration, and the formula of each evaluation index was as follows:

$$\text{RMSE} = \sqrt{\frac{1}{M} \sum_{m=1}^{M} (t_m - \hat{t}_m)^2} \tag{2}$$

$$\text{MAE} = \frac{1}{M} \sum_{m=1}^{M} \left| t_m - \hat{t}_m \right| \tag{3}$$

where $M$ is the number of datasets for the period of feeding behavior and drinking behavior; $m$ is the sample number of the datasets; $t_m$ is the actual feeding or drinking duration; and $\hat{t}_m$ is the measured value for the feeding or drinking duration.

### 2.5.1. VGG19

A VGG series convolutional neural network is a very deep CNN developed by the Visual Geometry Group of the University of Oxford. It has been applied in many agricultural studies and achieved good results [21], including the recognition of pigs' faces [15,22]. Compared with the previous VGG16 model, the VGG19 used in this study added three convolutional layers, which improved the depth of the model and could effectively improve the accuracy. The architecture of VGG19 is shown in Figure 7. The input size of the image is 224 × 224 pixels, and the model consists of 16 convolution layers (Conv) with a ReLU activation function, five max-pooling layers, and a global average pooling layer. Finally, two dense layers were used and output by the Softmax activation function.

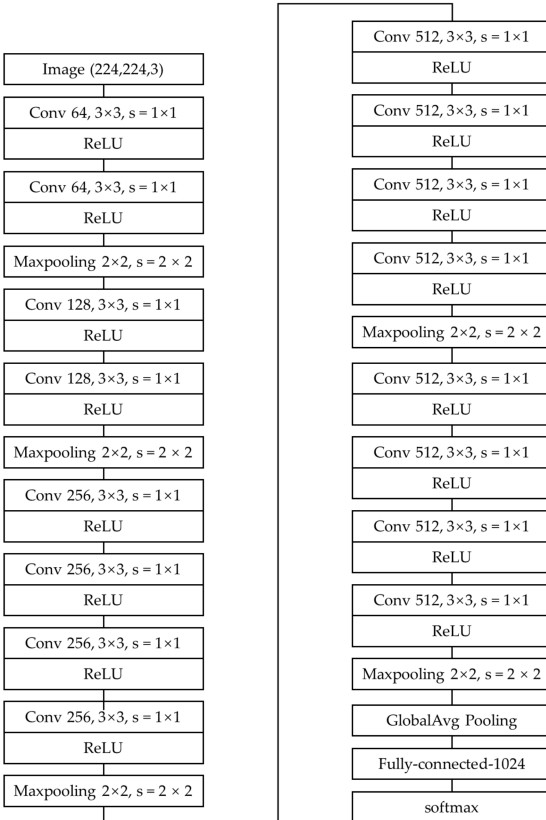

**Figure 7.** The VGG19 architecture.

### 2.5.2. Xception

Xception is an improved model of InceptionV3 [23] developed by Google, based on a linear stack of depth-wise separable convolution layers with residual connections [24]. Different from the VGG algorithm, which simply increases the number of convolution layers to improve the model efficiency, the use of depth-wise separable convolutions can reduce the computational complexity and make the model more easily defined and modified. The architecture of Xception is shown in Figure 8. The input size of the image is 299 × 299 pixels, and the model consists of 36 convolution layers, four max-pooling layers, and a global average pooling layer. Furthermore, 34 convolution layers were depth-wise separable convolution layers (Sep-Conv), and the above 36 convolutional layers were divided into 14 modules, all of which had linear residual connections around them, except the first and last modules. In addition, each convolution layer was followed by batch normalization (not marked in the figure). Finally, similar to the VGG19 model used in this study, ReLU was used as the activation function in the convolution layers, and two dense layers were used and output by the Softmax activation function.

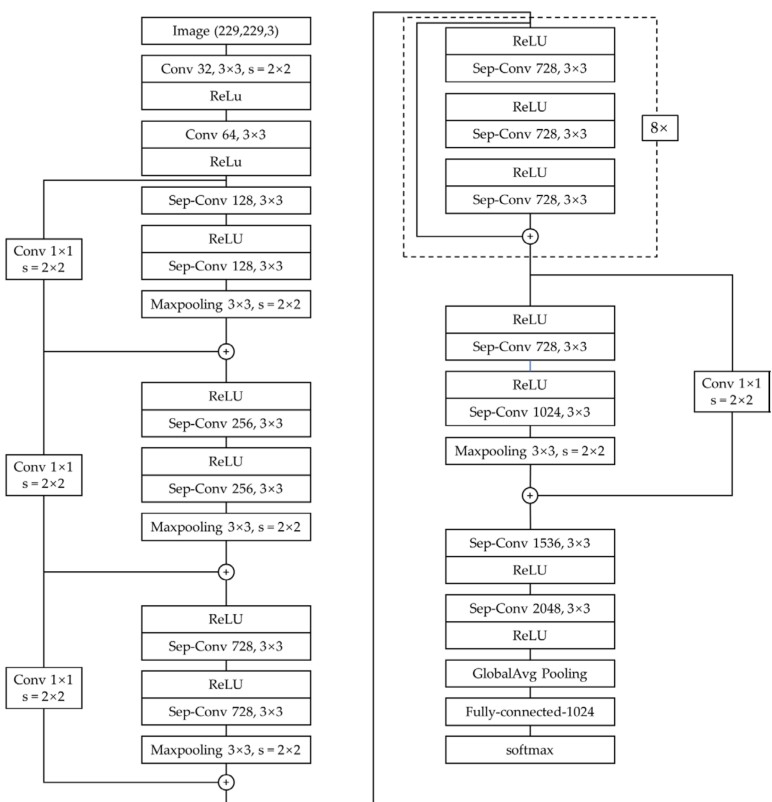

**Figure 8.** The Xception architecture.

### 2.5.3. MobileNetV2

MobileNetV2 is a lightweight CNN (Figure 9a) based on the bottleneck residual block and can effectively reduce the parameters of the model [25]. The input size of the image is 224 × 224 pixels, the initial and final convolutional layers are both fully convolutional layers, and there are 17 residual bottleneck layers divided into seven sequences in the middle. Each sequence repeats a different number of times (the repeat times of each sequence are shown in Figure 9a). The stride(s) marked in each sequence in Figure 9a represents the s in the first layer, and all other layers used s = 1 × 1. Figure 9b,c shows the residual bottleneck layer structure when s = 1 × 1 and s = 2 × 2, respectively. In addition, unlike VGG19 and Xception, ReLU6, which can improve the robustness of low-precision computation [26], was used as the activation function in the convolutional layers and residual bottleneck

layers. Finally, the model also had a global average pooling layer, and two dense layers were used and output by the Softmax activation function.

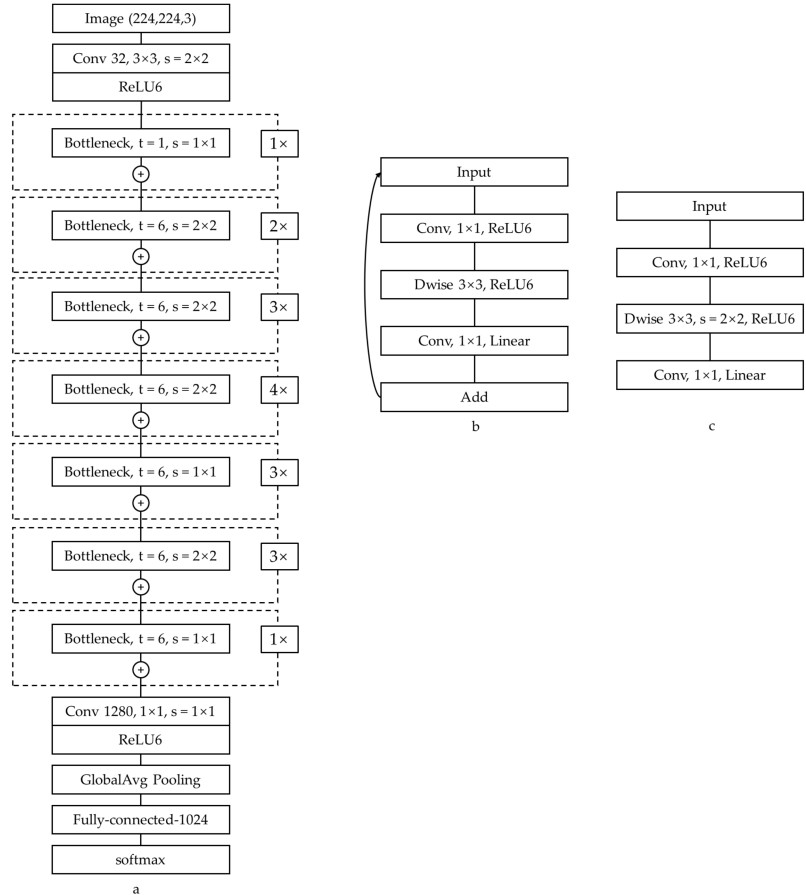

**Figure 9.** The MobileNetV2 architecture. Overall architecture (**a**); the residual bottleneck layer structure when s = 1 × 1 (**b**); the residual bottleneck layer structure when s = 2 × 2 (**c**).

## 3. Results and Discussion

### 3.1. Model Training

The model size and training duration for the training of recognition of feeding behavior and drinking behavior are shown in Table 1. The model trained by VGG19 had a medium size and training duration, with a training duration of 132 min 30 s (159 s per iteration) for the feeding dataset and 80 min 03 s (96 s per iteration) for the drinking dataset. The model trained by Xception was the largest and had the longest training duration of 275 min 59 s (331 s per iteration) and 190 min 19 s (228 s per iteration) for the feeding and drinking datasets, respectively. The model trained by MobileNetV2 was the smallest and had the shortest training duration of 56 min 26 s (68 s per iteration) for the feeding dataset and 31 min 53 s (38 s per iteration) for the drinking dataset.

**Table 1.** Model information.

| Model | Model Size (MB) | Training Duration |
| --- | --- | --- |
| VGG19—feeding | 241 | 132 min 30 s |
| Xception—feeding | 269 | 275 min 59 s |
| MobileNetV2—feeding | 42 | 56 min 26 s |
| VGG19—drinking | 241 | 80 min 03 s |
| Xception—drinking | 269 | 190 min 19 s |
| MobileNetV2—drinking | 42 | 31 min 53 s |

The results of each model are shown in Figure 10. The accuracy and loss of the models trained by MobileNetV2 were basically consistent with those of Xception and VGG19, with the accuracy of the validation dataset in FARS and DARS exceeding 99%. However, the results of VGG19 had great fluctuation in the model training of the feeding dataset, which might be because the architecture of VGG19 was just a simple stack of convolution layers and pooling layers. In addition, for the training of the feeding and drinking datasets, the accuracy of the models trained by Xception reached 98% in the first epoch, while VGG needed more than five epochs to achieve the same effect, and for MobilenetV2, more than seven epochs were needed. Considering the training duration of each iteration, although MobilenetV2 needed more epochs to reach a high accuracy, the whole training duration was the shortest. Therefore, in the training stage, MobilenetV2 achieved the best results, Xception was in second place, and the VGG model was the poorest.

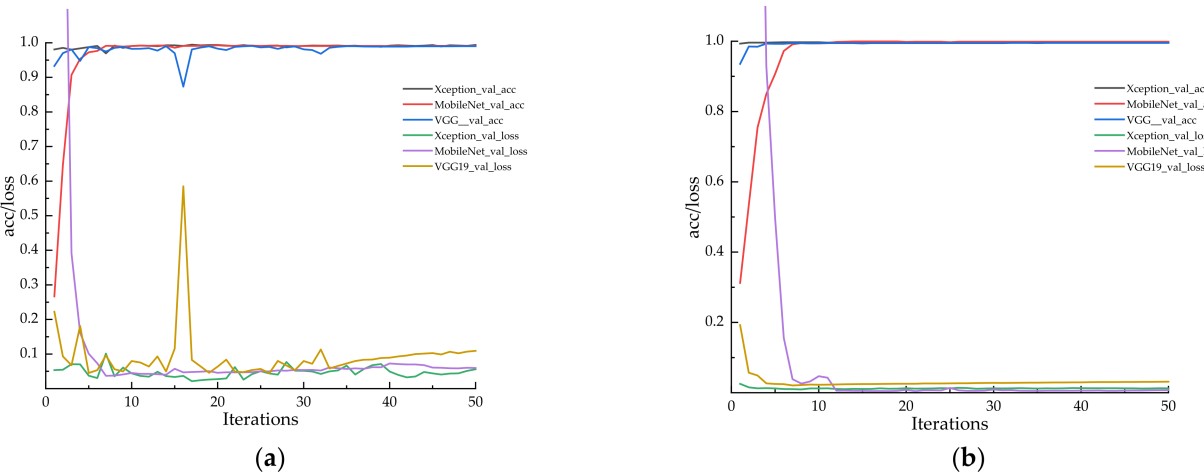

(**a**)                                             (**b**)

**Figure 10.** The training results of the VGG19, Xception, and MobilenetV2 models. Results of the feeding validation dataset (**a**); results of the drinking validation dataset (**b**).

### 3.2. Model Test

In this study, 130 sets of continuous images from DARS (2987 images) and 50 sets of continuous images from FARS (1809 images) were used for the test. For a shooting frequency of one image per second, the average drinking duration was 27.72 s, and the average feeding duration was 21.37 s. The performances of each model are shown in Tables 2 and 3.

**Table 2.** Recall rates of the models for the drinking test dataset.

| Item | VGG19 | MobileNetV2 | Xception |
|---|---|---|---|
| 0 | 100.00% | 100.00% | 100.00% |
| 1 | 98.71% | 98.33% | 98.20% |
| 2 | 98.58% | 99.82% | 98.76% |
| 3 | 99.32% | 100.00% | 99.78% |
| 4 | 97.98% | 100.00% | 100.00% |
| 5 | 94.31% | 97.28% | 94.80% |
| 6 | 99.44% | 99.62% | 99.44% |
| RMSE (s) | 0.86 | 0.60 | 0.81 |
| MAE (s) | 0.30 | 0.12 | 0.26 |

As shown in Table 2, for the testing dataset from DRAS, MobileNetV2 achieved the best performance with a recognition recall rate of each pig higher than 97%, and the recall rate of pigs 3 and 4 was 100%. In addition, the model trained by MobileNetV2 was used to estimate the drinking duration, and RMSE and MAE were only 0.60 s and 0.12 s. The effects of the models trained by VGG19 and Xception were similar, which showed that, except for the recognition recall rate of pig 5, which was lower than 95%, the recognition

recall rate of other pigs was higher than 98%. It is worth mentioning that the recall rate of all the models for distinguishing between drinking and nondrinking was 100%.

**Table 3.** Recall rates of the models for the feeding test dataset.

| Item | VGG19 | MobileNetV2 | Xception |
|---|---|---|---|
| 0 | 100.00% | 100.00% | 100.00% |
| 1 | 100.00% | 99.57% | 99.57% |
| 2 | 99.78% | 99.78% | 99.55% |
| 3 | 100.00% | 99.47% | 100.00% |
| 4 | 97.08% | 97.92% | 93.75% |
| 5 | 100.00% | 100.00% | 99.26% |
| 6 | 98.58% | 98.58% | 97.16% |
| RMSE (s) | 0.62 | 0.58 | 1.38 |
| MAE (s) | 0.21 | 0.21 | 0.49 |

The performance of the testing dataset from FARS was also very good (Table 3): all models distinguished between feeding and nonfeeding 100% of the time. It is worth mentioning that VGG19, which had the poorest effect in the model training stage, achieved the best performance in the testing stage. The recall rate for individual pig recognition exceeded that of MobileNetV2, with a 100% recall rate for identifying pigs 1, 3, and 5. However, because of the differences in the feeding habits of pigs, the image number of each pig was not the same. Therefore, the RMSE of VGG19 was higher than that of MobilenetV2 for estimating the feeding duration. Unexpectedly, Xception, which achieved good results in the training stage and could reach a high level with only one epoch, did not achieve good performance in the testing stage; the recognition recall rate was only 93.75% for pig 4.

As a lightweight model, MobileNetV2 achieved the best performance with the lowest training duration. Compared with previous studies, the model trained by MobileNetV2 in this study had significantly improved the accuracy of individual pig recognition, and the model training process was simpler, without complex model training operations. These findings could be attributed to FARS and DARS limiting the image shooting area, reducing the impact of a complex background environment on the recognition of pigs, and a lightweight model could be used in recognizing pigs. However, MobilenetV2 was a lightweight model after all. Whether it could achieve the same efficiency of individual recognition when there were more pigs remains to be further studied. VGG19, which was introduced many years ago [21], still achieved good performance. While, due to its large fluctuations and long training duration, VGG19 might cause a series of problems in subsequent applications, the VGG19 was not suitable to be promoted and applied as the core algorithm in feeding and drinking behaviors recognition. As a model with many excellent results [16,18,27,28], although Xception did not achieve the best results in this study, it still got a good recall rate in recognizing most pigs. Because the FARS and DARS could effectively reduce the impact of the background, the advantages of the Xception might be difficult to capture, the Xception model might be more suitable for building complex models or situations, where the computing power was high enough.

### 3.3. Application Prospects

In view of the excellent achievements of MobilenetV2, in this study, models trained by MobileNetV2 were chosen for building the application. LabVIEW and its Python Integration Toolkit were combined to design a new behavior monitoring software to monitor the feeding and drinking behaviors of pigs, which could automatically be used for FARS and DARS.

The model trained by MobilenetV2 was able to distinguish whether pigs were feeding/drinking. Thus, in this software, when the software recognized that a pig was feeding or drinking for 1 s, the system captured all of the pictures taken subsequently until it was determined that the pig had not been feeding or drinking for 1 s. Then, all images taken by this system were recognized, the pig with the highest number of recognition results was

the final recognition result, and the number of images was the duration of this feeding or drinking behavior. The pig's feeding or drinking duration was displayed on the front panel (Figure 11) and saved in the database in real time. The operator can change the display content into a daily feeding or drinking duration according to the user's needs.

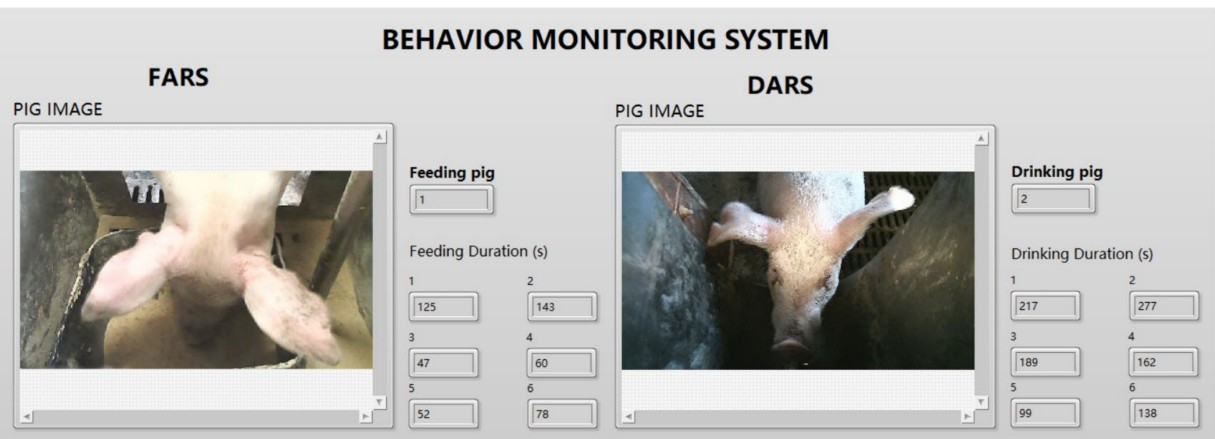

**Figure 11.** Front panel of the behavior monitoring system.

The behavior monitoring software based on LabVIEW in this study could be combined with FARS or DARS and applied to group-housed pigs, singly housed pigs, EFS, and FIRE, regardless of feeding conditions. Users can mark the abnormal pigs by recording their individual feeding and drinking behaviors, adjusting the feeding plan in time, and improving the feed utilization rate in combination with the feed line and other equipment. Furthermore, compared with the one-time RFID tags, the camera has a longer service life and is easier to maintain and replace, which means that the system can effectively reduce the cost of ear tags and labor input. In addition, the application of this study was simple, with strong reproducibility. The model training process was relatively simple, and the training speed was fast. LabVIEW is an unusual graphical programming language based on virtual instruments, which can assist nonprogrammer users and researchers in using and reproducing this system [29]. Producers can operate the system only if they have basic computer knowledge. In case of program failure, developers can also debug the program through a remote desktop.

## 4. Conclusions

In this study, two systems named FARS and DARS were designed to monitor the individual drinking and feeding behaviors of pigs, and a systematic application design was carried out to facilitate their better application in two systems in the future. Specifically, the best model trained by MobileNetV2 achieved good performance. The recall rate of recognizing six pigs was higher than that in previous research with the recall rate higher than 97%. In addition, its estimated RMSE and MAE for drinking duration were 0.60 s and 0.12 s, respectively, and the estimated RMSE and MAE for feeding duration were 0.58 s and 0.21 s, respectively. This method can be applied to group-housed pigs, singly housed pigs, pig insurance registration, pig individual status monitoring, and other occasions with strict requirements for individual identification. Only a small amount of pig data needs to be collected and corrected on the basis of the original mode output.

**Author Contributions:** Data curation, Y.Z., K.Z. and Z.Z.; investigation, G.T.; methodology, Y.Z. and K.Z.; project administration, G.T.; software, Y.Z.; supervision, G.T.; writing—original draft, Y.Z.; writing—review and editing, K.Z., Z.Z., H.J. and G.T. All authors have read and agreed to the published version of the manuscript.

**Funding:** This work was funded by the Special Key Project of Chongqing Technology Innovation and Application Development (cstc2021jscx-dxwtBX0006).

**Institutional Review Board Statement:** Not applicable.

**Informed Consent Statement:** Not applicable.

**Data Availability Statement:** All datasets from the current study are available from the corresponding author upon reasonable request.

**Conflicts of Interest:** The authors declare no conflict of interest.

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
