# Peer review of "Systems to Monitor the Individual Feeding and Drinking Behaviors of Growing Pigs Based on Machine Vision"

_agriculture, doi:10.3390/agriculture13010103_

Round 1

Reviewer 1 Report

Dear authors,

The three models were efficient according to their results; however, I attach general comments to the manuscript that the authors should address to improve the paper.

- It is not very clear to me how this would help the producers (what benefits would incorporating this system bring to their stables?)

- Continuing with the above, what does it cost to implement this system, how long and how difficult would it be for a producer to learn to interpret this data, (would courses be given?

-The statistical analysis is not mentioned and there is no comparison between the models, the authors must explain this section.

Reviewer 2 Report

GENERAL COMMENT: The authors submitted a manuscript identifying swine behaviour using image analysis. The designed neural network and other machine vision algorithms are very actualised and with promising results. The article is very well written and easy to follow. Only some suggestions for minor mistakes: In the Introduction, please avoid older citations. Also, the other citations must be actualised based on the novelty of this study. The conclusion must be reduced without loss of quality. Finally, I'd be delighted if the authors provided the code as supplementary material.

Round 2

Reviewer 1 Report

Dear Authors,

The authors attended to the comments, they should only correct the style of the English.